# The prevalence and correlates of obstructive lung disease among adults aged 45 and above in India: Findings from the longitudinal aging study in India

T. V. Sekher[1☉], Emma Nichols[2,3☉*], Sundeep Salvi[4,5], Sarang P. Pedgaonkar[1], Crystal M. North[6,7], Sarah Petrosyan[2], David E. Bloom[8], Jinkook Lee[2,9]

**1** International Institute for Population Sciences, Mumbai, Maharashtra, India, **2** Center for Economic and Social Research, University of Southern California, Los Angeles, California, United States of America, **3** Leonard Davis School of Gerontology, University of Southern California, Los Angeles, California, United States of America, **4** Pulmocare Research and Education Foundation (PURE), Pune, Maharashtra, India, **5** School of Health Sciences, Symbiosis International Deemed University, Pune, Maharashtra, India, **6** Division of Pulmonary and Critical Care Medicine, Department of Medicine, Massachusetts General Hospital, Boston, Massachusetts, United States of America, **7** Medical Practice Evaluation Center, Massachusetts General Hospital, Boston, Massachusetts, United States of America, **8** Department of Global Health and Population Research, Harvard T.H. Chan School of Public Health, Boston, Massachusetts, United States of America, **9** Department of Economics, University of Southern California, Los Angeles, California, United States of America

☉ These authors contributed equally to this work.
* emmanich@usc.edu

**Data availability statement:** Longitudinal Aging Study in India (LASI) data used in this study, with the exception of spirometry data, are available after completion of a signed data access agreement form on the websites of the

## Abstract

Despite its high disease burden, few existing estimates of the prevalence of obstructive lung disease in India are based on high-quality data sources. To our knowledge, prior studies at the national level have not included objective measurements of lung function. The Longitudinal Aging Study in India administered spirometry to adults 45 years and older (N = 31,103). We estimated inverse probability weights to account for sample selection processes and quantified the prevalence of obstructive lung disease overall and by region, age, and gender. We investigated the overlap among objective rates of obstructive lung disease, respiratory symptoms, and self-reported diagnoses. Additionally, we evaluated associations between obstructive lung disease and pertinent risk factors. The overall prevalence of obstructive lung disease was 14.4% (95% confidence interval [CI] 13.4–15.4). Prevalence was higher among men than women (p < 0.001) and increased with age (p < 0.001). Disease awareness was low, with only 12.0% (95% CI 9.9–14.5) of men and 11.0% (95% CI 8.6–14.0) of women with obstructive lung disease reporting prior diagnoses of lung disease. We observed heterogeneity by region (p < 0.001), which largely remained after accounting for differences in demographic and risk factors. High prevalence and low disease awareness highlight important challenges in the prevention and management of obstructive lung disease in India. Multifaceted approaches are needed to address this disease



Gateway to Global Aging Data (https://g2aging.org/survey/get_data) and the International Institute for Population Sciences (https://www.iipsindia.ac.in/content/LASI-data). Spirometry data will be made available to researchers by request, sent to the LASI Data Access Team (contact via help@lasi-dad.org or chiayisc@usc.edu). Registration and data access forms are required under the Institutional Review Board's human subjects research policies and requirements.

**Funding:** Data collection was funded by the Ministry of Health and Family Welfare, Government of India [T22011/02/2015-NCD to TVS, DEB, and JL] and the National Institute on Aging, National Institutes of Health (NIA/NIH), USA [R01AG042778 to DEB, JL, and TVS]. The preparation of this article was funded by the NIA/NIH [R01AG030153 to EN and JL]. The funders had no role in study design, data collection and analysis, decision to publish, or preparation of the manuscript.

burden, including understanding and lowering exposure to risk factors and improving infrastructure and resources for diagnosing and managing obstructive lung disease.

## Introduction

Obstructive lung disease is a leading cause of morbidity and mortality globally [1,2]. More specifically, estimates suggest that 90% of deaths due to chronic obstructive pulmonary disease (COPD), which is the primary cause of obstructive lung disease in older adults, occur in lower- and middle-income countries [3]. Challenges related to high levels of morbidity, chronic symptoms, activity limitations, and ongoing disease exacerbations are compounded in lower- and middle-income countries, where limited resources are available for disease prevention and management [4].

In India, the world's most populous country, the overall burden of obstructive lung disease is increasing and represents a growing proportion of health loss from all diseases [5]. Existing studies have reported prevalences of COPD in India ranging from 4.2% to 8.0% [5–7]. Although these studies present national estimates, contributing sources with primary data from objective spirometry are limited to studies conducted in restricted geographic areas. Global Burden of Disease (GBD) 2016 study estimates primarily rely on data from the Burden of Obstructive Lung Disease study group in Delhi, Kashmir, Pune, and Mumbai, as well as smaller localized studies [8–10]. Although the GBD study group uses information on causes of death, covariates, and a small body of available hospital records to further inform estimates, ultimately the quality of evidence is limited by the small number of studies on prevalence [5]. A recent meta-analysis (2021) also synthesized evidence from eight studies with objective spirometry assessments in India [11]; however, these studies focused either on specific cities or populations (e.g., premenopausal women exposed to biomass smoke) [11].

Although national estimates of the prevalence of COPD or obstructive lung disease exist, the quality of underlying data sources compromises the reliability of estimates [12]. Furthermore, national-level data obtained using consistent methods are needed to enable unbiased comparisons across different regions of India, which is of particular importance considering known regional heterogeneity in contextual and environmental factors and other risks expected to affect disease prevalence. Given the limitations of existing evidence, a crucial need exists for national data on obstructive lung disease based on objective spirometry to improve existing prevalence estimates, examine regional variation, and assess associations with risk factors.

The Longitudinal Aging Study in India (LASI) is, to our knowledge, the first study to administer objective spirometry assessments to a nationally representative sample of adults (over 45 years of age) in India. Here, we report on the national and regional prevalence of obstructive lung disease in India and examine self-reported symptoms and disease awareness. Given that most evidence on risk factors for obstructive lung disease comes from high-income contexts and that results may not be generalizable,



we also investigate associations with demographic and risk factors such as gender, socioeconomic status, and unclean cooking fuel use.

## Materials and methods

### Sample

LASI is a nationally representative sample of more than 73,000 adults 45 years and older and their spouses [13]. Participants were recruited and data was collected between April 1, 2017, and October 20, 2021, due to phased data collection efforts. Except for data from the state of Sikkim, all data collection was completed from 2017−18. All participants provided informed consent (written or thumbprint), and study protocols were approved by the relevant institutional review boards at participating institutions.

The current analysis was limited to participants 45 years of age and older who consented to physical biomarker data collection (response rate: 87.9%) [14]. We used sampling weights for the biomarker sample to account for the overall sampling scheme and to ensure that the demographic characteristics of the biomarker sample were in line with the reference sample [15]. The biomarker sample included a total of 60,591 individuals over 45 years of age.

### Spirometry data collection and processing

The collection of spirometry data was part of the physical biomarkers module but required additional screening and consent. Contraindications included current tuberculosis treatment; current upper respiratory infection; pregnancy, chest or abdominal surgery in the past three months; heart attack or hospitalization for heart problems in the last three months; and eye surgery in the last three months. Ninety-one percent of eligible individuals consented to spirometry.

Interviewers underwent a 5-day rigorous and standardized training protocol. Following training, interviewers measured forced expiratory volume in one second ($FEV_1$) and forced vital capacity (FVC) with Thor handheld ultrasonic spirometers (Thor Medical Systems, Budapest, Hungary). Field investigators performed biometric calibrations at least once every 15 days. Spirometry was conducted using the American Thoracic Society/European Respiratory Society 2005 guidelines. Interviewers were instructed to perform three acceptable trials, of which two needed to also be repeatable, but not perform more than eight trials per individual (at least 30 seconds between trials). Spirometry was stopped early if the respondent coughed continuously, felt lightheaded, or was unable to perform the test.

Spirometry experts from the Chest Research Foundation, Pune, reviewed the data for American Thoracic Society/European Respiratory Society acceptability. Given that LASI is a community-based study and spirometry was conducted in respondents' homes rather than ideal laboratory settings, guidelines were slightly modified to require at least one acceptable spirometry trial, with a forced expiratory time of at least three seconds or with at least a one second plateau. Of respondents with spirometry measurements, 62% fulfilled the acceptability criteria. Among those with acceptable spirometry, $FEV_1$ was repeatable for 98% and FVC was repeatable for 97% of respondents, and differences in spirometry parameters were not meaningfully different between those with and without repeatable trials after adjusting for age and gender (difference of 0.04 L for $FEV_1$, 0.08 L for FVC). The final analytic sample included N = 31,103 respondents (S1 Fig). To account for potential selection bias due to lack of consent or unacceptable spirometry measurements, we used inverse probability weights (see below).

Obstructive lung disease was defined as $FEV_1/FVC < 70\%$ [16]. Bronchodilation was not included in the LASI protocol due to feasibility issues; therefore, we interpret findings as indicative of obstructive lung disease generally rather than as COPD. We used the Global Lung Function Initiative 2012 prediction equation to define severity categories [17]. Absent Indian-specific parameters, we used Southeast Asian parameters as the closest approximation. Mild, moderate, and severe/very severe obstructive lung disease were defined based on the percent predicted $FEV_1$ (mild: ≥ 80%; moderate: ≥ 50% to <80%; severe and very severe: < 50%) [16].



### Other measured variables

Covariates included respondents' self-reported age, gender, education, literacy, marital status, caste, smoking status, frequency of moderate and vigorous physical activity, use of unclean cooking fuel, and self-reported tuberculosis in the last two years. We also considered the number of difficulties reported on activities of daily living and instrumental activities of daily living. Residence in a rural or urban setting was based on the 2011 Indian Census. Body mass index (BMI) was calculated based on measured height and weight and categorized using Indian-specific thresholds [18]. Using World Health Organization thresholds, distributions were different but associations were similar (S1 and S2 Tables). Details on covariates are available in S1 File.

We assessed self-reported symptoms of lung disease within the past two years, including cough, dizziness, shortness of breath, and wheezing. Additionally, we considered self-reported diagnoses of lung disease (COPD, asthma, or chronic bronchitis).

### Construction of weights to account for selection bias

We used three logistic regression models to model 1) contraindications, 2) lack of consent to spirometry, and 3) not meeting spirometry acceptability criteria. We used covariates representing key demographic characteristics, overall health, and respiratory symptoms (S2 File). We estimated stabilized inverse probability weights and confirmed that standardized mean differences between the weighted and full sample were small (S2 Fig). We multiplied the three weights and the biomarker weight together to calculate weights considering unequal sampling probabilities and all selection processes.

### Statistical analysis

We first assessed the characteristics of the weighted sample overall and by region (S3 Fig). We then compared the distributions for the $FEV_1/FVC$ ratio by region and calculated the overall and region-specific prevalence of obstructive lung disease, the prevalence of each severity category, and the prevalence stratified by gender and age group. We examined the proportion of participants who reported respiratory symptoms (cough, dizziness, shortness of breath, wheezing) and who self-reported diagnoses of lung diseases (asthma, COPD, chronic bronchitis) among those with and without obstructive lung disease based on objective spirometry performance, stratified by gender. We used logistic regression models adjusted for age and stratified by gender to evaluate the statistical significance of differences. We compared the regional distribution of objective prevalence, self-reported prevalence, and disease awareness.

We also evaluated associations between obstructive lung disease and various factors, including demographic characteristics (age, gender), socioeconomic markers (rural/urban residence, caste, education), and risk factors (BMI category, smoking status, unclean cooking fuel use, tuberculosis). We used Poisson regression with robust standard errors to estimate prevalence ratios. We estimated unadjusted prevalence ratios; prevalence ratios adjusted for age and gender (model 1); and prevalence ratios adjusted for age, gender, and smoking status (model 2). To understand whether differences among demographic characteristics or risk factors could explain observed differences among regions, we compared estimates from an unadjusted model with indicator variables for region with estimates from models additionally adjusted for 1) age and gender; 2) age, gender, and smoking status; and 3) age, gender, and all included risk factors. In regression models, we excluded all respondents with missing data on any of the characteristics of interest (N = 415) (S1 Fig). All analyses accounted for survey and selection weights as well as the complex sampling design (primary sampling units and strata). All analyses were conducted in R version 4.2.2.

### Results

The weighted analytic sample (N = 31,103) had an average age of 59.1 years with an approximately even gender distribution (47.3% women) (Table 1). About half of the sample had no education (52.9%), though the proportion with no

**Table 1. Demographic characteristics of the included participants from the Longitudinal Aging Study in India (N = 31,103) overall and by region. Characteristics are weighted to account for differential sampling probabilities and selection into the spirometry sample. Means and standard deviations are shown for continuous variables (age); proportions and numbers are shown for binary and categorical variables (all other variables). Square brackets show the number of missing records for binary or continuous variables. BMI refers to body mass index.**

| | Overall (N = 31103) | North (N = 5372) | Central (N = 2582) | East (N = 7237) | Northeast (N = 6348) | West (N = 3662) | South (N = 5902) |
|---|---|---|---|---|---|---|---|
| Women | 47.3 (15238) | 49.1 (2733) | 46.7 (1105) | 45.0 (3571) | 42.2 (3025) | 48.1 (1861) | 48.8 (2943) |
| Age | 59.1 (10.0) | 59.4 (10.3) | 59.9 (10.3) | 58.4 (9.7) | 57.2 (9.2) | 59.1 (9.9) | 59.1 (9.7) |
| 45–54 | 37.4 (13276) | 36.5 (2199) | 34.2 (1043) | 40.1 (2945) | 44.6 (2890) | 37.5 (1528) | 37.7 (2671) |
| 55–64 | 32.8 (10056) | 32.6 (1740) | 34.0 (916) | 33.7 (2419) | 32.6 (1925) | 32.3 (1171) | 31.4 (1885) |
| 65–74 | 21.4 (6023) | 20.1 (1093) | 22.2 (490) | 19.4 (1437) | 17.4 (1117) | 22.3 (793) | 22.9 (1093) |
| 75+ | 8.4 (1748) | 10.8 (340) | 9.7 (133) | 6.8 (436) | 5.4 (416) | 8.0 (170) | 8.1 (253) |
| Rural | 68.5 (20804) | 63.5 (3427) | 75.7 (2000) | 75.9 (5205) | 77.6 (4755) | 58.4 (1921) | 63.8 (3496) |
| Caste | | | | | | | |
| No caste or other caste | 26.4 (8836) | 39.7 (2758) | 22.1 (605) | 32.5 (2317) | 29.8 (1044) | 35.4 (1281) | 12.1 (831) |
| Scheduled tribe | 20.0 (5257) | 23.7 (1308) | 24.1 (574) | 22.9 (1447) | 10.2 (532) | 15.9 (470) | 15.9 (926) |
| Scheduled caste | 8.4 (6081) | 8.4 (296) | 8.2 (228) | 9.3 (894) | 29.5 (3487) | 11.4 (615) | 3.6 (561) |
| Other backward class | 44.8 (10631) | 28.1 (1003) | 45.6 (1174) | 34.1 (2500) | 30.3 (1275) | 37.2 (1275) | 68.2 (3404) |
| Missing | 0.3 (298) | 0.1 (7) | 0.0 (1) | 1.2 (79) | 0.2 (10) | 0.1 (21) | 0.2 (180) |
| Education | | | | | | | |
| No school | 52.9 (13089) | 54.8 (2299) | 61.8 (1320) | 54.2 (3405) | 46.0 (2434) | 43.1 (1258) | 49.7 (2373) |
| Less than secondary school | 21.4 (8045) | 14.2 (1118) | 15.1 (516) | 22.1 (1825) | 26.2 (1812) | 29.2 (1176) | 24.8 (1598) |
| Secondary school and higher | 25.7 (9969) | 30.9 (1955) | 23.1 (746) | 23.7 (2007) | 27.8 (2102) | 27.7 (1228) | 25.6 (1931) |
| BMI category | | | | | | | |
| Normal | 16.4 (3678) | 11.8 (411) | 24.8 (499) | 19.8 (1319) | 17.4 (684) | 12.6 (368) | 10.7 (397) |
| Underweight | 41.4 (12899) | 36.6 (1824) | 45.8 (1257) | 47.2 (3373) | 51.5 (3103) | 38.2 (1326) | 35.9 (2016) |
| Overweight | 14.5 (4914) | 16.6 (895) | 10.9 (308) | 13.8 (1017) | 13.8 (1064) | 15.6 (598) | 16.6 (1032) |
| Obese | 26.5 (9522) | 33.9 (2226) | 17.4 (512) | 18.7 (1509) | 17.1 (1479) | 32.9 (1363) | 34.2 (2433) |
| Missing | 1.3 (90) | 1.1 (16) | 1.1 (6) | 0.5 (19) | 0.2 (18) | 0.7 (7) | 2.6 (24) |
| Smoking (current or former) | | | | | | | |
| Never | 79.4 (25173) | 70.2 (3992) | 77.7 (2008) | 80.4 (6070) | 78.9 (4853) | 86.0 (3260) | 81.0 (4990) |
| Current | 4.2 (1396) | 5.3 (315) | 3.0 (76) | 4.6 (286) | 5.7 (399) | 2.8 (111) | 4.9 (209) |
| Former | 16.4 (4488) | 24.3 (1046) | 19.2 (497) | 15.0 (878) | 15.2 (1083) | 11.1 (288) | 14.0 (696) |
| Missing | 0.1 (46) | 0.2 (19) | 0.1 (1) | 0.1 (3) | 0.2 (13) | 0.1 (3) | 0.1 (7) |
| Unclean cooking fuel use | 47.4 (13969) [369] | 43.7 (1973) [81] | 60.1 (1509) [24] | 64.9 (4500) [23] | 65.4 (3594) [39] | 34.9 (1033) [99] | 28.6 (1360) [103] |
| Tuberculosis (self-report, last 2 years) | 0.7 (185) [4] | 0.6 (36) [1] | 0.8 (17) [0] | 0.6 (33) [1] | 0.7 (46) [0] | 1.2 (29) [0] | 0.2 (24) [2] |

education varied by region and was highest in the Central (61.8%), Northern (54.8%), and Eastern (54.2%) regions. The prevalence of risk factors for obstructive lung disease also varied by region. The prevalence of smoking (current or former) ranged from 29.6% in Northern India to 13.9% in Western India, and the prevalence of unclean cooking fuel use also varied, with the lowest observed prevalence (28.6%) in Southern India and the highest observed prevalence (65.4%) in Northeast India.

We estimated the overall prevalence of obstructive lung disease in India to be 14.4% (95% confidence interval [CI] 13.4–15.4) (Fig 1; S3 Table). Although the shape of the distributions of $FEV_1$/FVC ratios differed somewhat, the proportion

of individuals with obstructive lung disease below the 70% threshold was less discrepant across regions. We observed the highest prevalence in Northern India (17.2%; 95% CI 15.0–19.6), followed by Western India (16.2%; 95% CI 13.8–18.9). Northeast India had the lowest observed prevalence (10.8%; 95% CI 8.2–14.0). When divided into severity categories, the largest category was moderate severity, followed by severe or very severe, and then mild. The prevalence of obstructive lung disease was higher in older age groups for both men and women, although the prevalence was higher in men compared with women for each age group (Fig 2; S4 Table). In men, the prevalence of obstructive lung disease among 45–54-year-olds was 10.6% (95% CI 9.4–11.9), approximately one-third of the prevalence estimate for men 75 years and older (32.2%; 95% CI 27.3–37.4). In comparison, an approximately two-fold difference in prevalence across age groups appeared for women, with an estimated prevalence of 7.9% (95% CI 6.8–9.2) in 45–54-year-olds and 18.1% (95% CI 12.5–25.4) in those 75 years and older. The distribution of disease severity among individuals with obstructive lung disease was more consistent across age and gender.

Respiratory symptoms were more common among those with obstructive lung disease based on spirometry (Fig 3); results were statistically significant for cough and shortness of breath (only in men), but not for dizziness or wheezing. Those with self-reported diagnoses of lung disease were also more likely to have objective obstructive lung disease, though disease awareness was low (Fig 3). Only 12.0% (95% CI 9.9–14.5) of men and 11.0% (95% CI 8.6–14.0) of women with objective obstructive lung disease based on spirometry self-reported any prior diagnosis of lung disease, including COPD, chronic bronchitis, or asthma. Though the prevalence of objective obstructive lung disease was highest in Northern India, the prevalence of self-reported lung disease was highest in Southern India, and disease awareness was

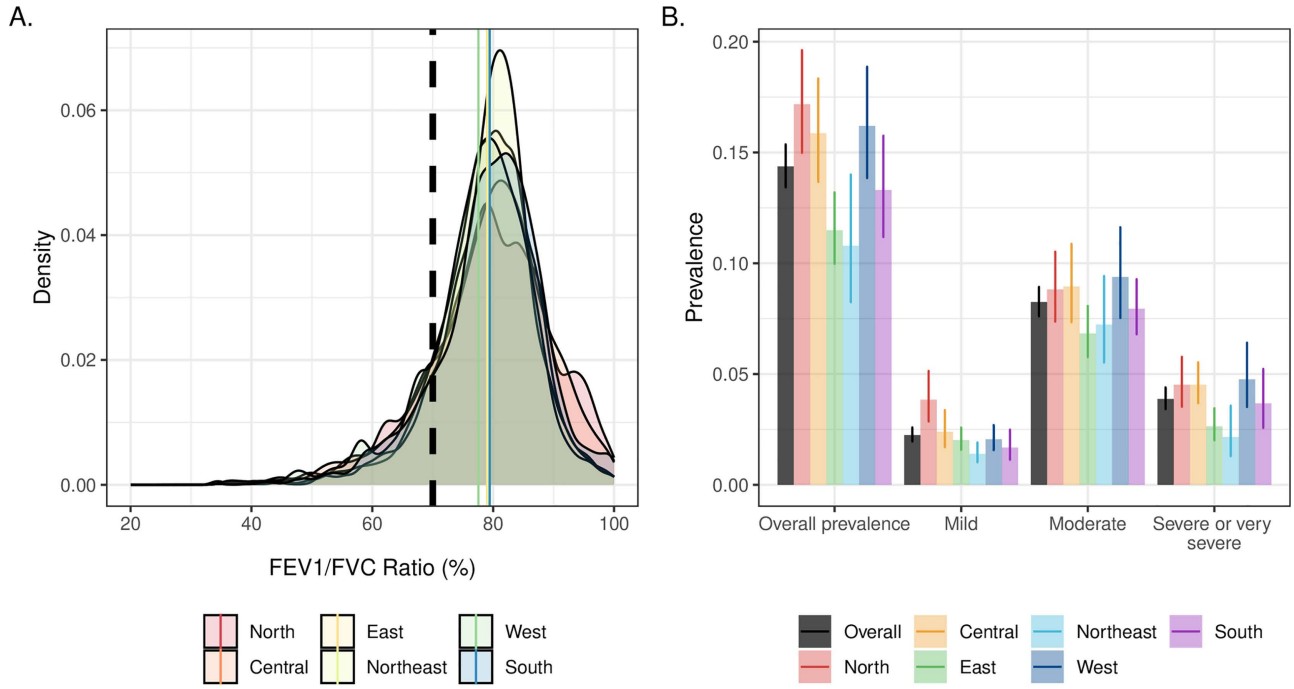

**Fig 1. Distributions of the forced expiratory volume in one second/forced vital capacity (FEV$_1$/FVC) ratio across geographic regions [A] and the prevalence of obstructive lung disease nationally and by region and overall and by severity in the Longitudinal Aging Study in India (N = 31,103) [B].** Vertical lines in Panel [A] show the mean FEV1/FVC ratio observed in each region compared with the threshold used to define obstructive lung disease (70%, represented by the dashed black vertical line). Error bars in Panel [B] show 95% confidence intervals. Density plots and prevalences are weighted to account for unequal sampling probabilities and selection into the spirometry sample.

lowest in the Northeast region (S5 Table; S4 Fig). Those with previously diagnosed lung disease had a higher prevalence of respiratory symptoms (Fig 3).

In unadjusted models, women, those with the highest educational attainment (secondary school and higher), and those with overweight or obese BMI were at lower risk of obstructive lung disease. In models controlling for age, gender, and smoking status, estimates were attenuated but remained statistically significant (Table 2). For example, estimates for gender suggest that the prevalence of obstructive lung disease was 26% (95% CI 17–34) lower among women than among men, adjusting for differences in age and smoking status. In contrast, older age, scheduled caste (compared with no caste or other caste), underweight BMI, smoking, unclean fuel use, and self-reported tuberculosis were associated with a higher risk of obstructive lung disease. Estimates for unclean fuel use were marginally significant after adjusting for age and gender (prevalence ratio: 1.09; 95% CI 0.98–1.22), but estimates were attenuated and not statistically significant after additionally adjusting for smoking status (prevalence ratio: 1.03; 95% CI 0.92–1.15). Risk factors explained only a small amount of regional heterogeneity (S6 Table).

## Discussion

Based on a nationally representative sample of adults aged 45 and above in India with objective spirometry data, we estimated that the prevalence of obstructive lung disease in adults 45 years and older was 14.4% (95% CI 13.4–15.4). We observed heterogeneity across regions of India, with the highest observed prevalences in Northern, Western, and Central India and the lowest observed prevalences in Northeast and Eastern India. Symptoms of lung disease and self-reported diagnoses were associated with objective obstructive lung disease, though disease awareness was low, with only 12.0% (95% CI 9.9–14.5) of men and 11.0% (95% CI 8.6–14.0) of women with objective obstructive lung disease reporting prior diagnoses. We observed the strongest associations between obstructive lung disease and age, gender, and smoking status, though we also saw some evidence for associations with educational attainment and BMI.

Our prevalence estimates are higher than the estimated 7.4% [6] or 7.0% [7] prevalence of COPD from the two previous meta-analyses in India, though differences are likely largely attributable to differences in the age distributions of included studies. While both meta-analyses included studies with adults as young as 25 or 30, the LASI sample only includes adults 45 years and older; because the prevalence of obstructive lung disease increases with age, a higher prevalence of obstructive lung disease would be expected in the LASI sample. Additionally, there are other causes of obstructive lung disease beyond COPD (the largest being asthma), which we were unable to exclude due to the lack of post-bronchodilator testing, though COPD does comprise most of the obstructive lung disease category in older adults. Updated data from the GBD 2019 study yields estimates of 9.6% and 13.8% for COPD and the larger chronic respiratory disease category in adults 45 years and older [19]. Our estimate of 14.4% (95% CI 13.4–15.4) is fairly consistent with the previously reported GBD estimate of 13.8% for all chronic respiratory diseases.

Our estimates of geographic heterogeneity across India are also somewhat aligned with prior findings from the GBD study, with both studies finding higher prevalence in Northern and Western India and lower prevalences in Eastern and Northeastern India [5]. However, GBD estimates found that the prevalence of COPD was fairly high in in Southern India compared with other regions, while this was not the case in our data (Southern India had the fourth highest prevalence out of the six regions) [5]. Differences could potentially stem from the fact that representative data from the Burden of Obstructive Lung Disease studies were not available for areas in Southern India; therefore, GBD estimates relied on information from smaller studies, such as one study among nonsmoking rural women in the Tiruvallur district of Tamil Nadu [10]. Selected samples in these studies may lead to bias in overall estimates. General similarities between estimates illustrate the validity of our data, while the existence of some differences highlights the ways in which our data can be used to improve estimates from meta-analyses or modeling efforts such as the GBD study.

We found that the awareness of lung disease in this population was low; less than 13% of both men and women with objective obstructive lung disease reported a prior diagnosis of lung disease. Prior work has highlighted issues around the

A.

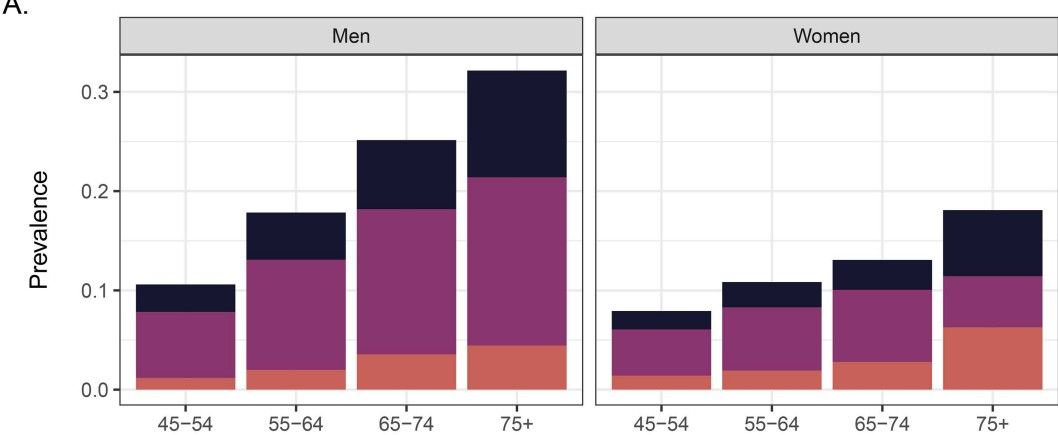

B.

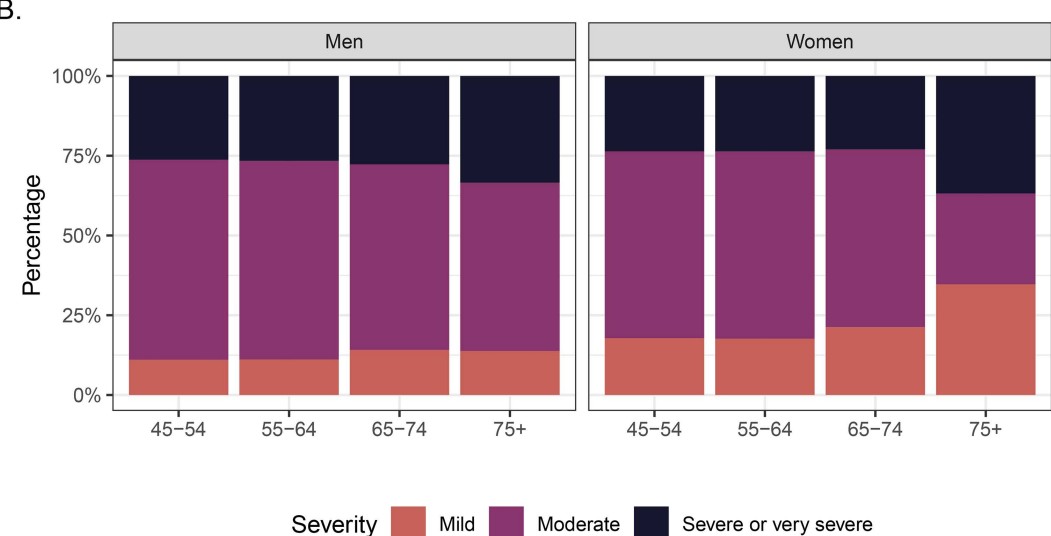

Severity: Mild — Moderate — Severe or very severe

**Fig 2. Prevalence of obstructive lung disease and severity category by age group and gender in the Longitudinal Aging Study in India (N = 31,103).** Panel [A] shows prevalences, and Panel [B] shows the proportion of cases in each severity category by age group and gender. Prevalences and proportions are weighted to account for unequal sampling probabilities and selection into the spirometry sample.

underdiagnosis of COPD in India due to low disease awareness [20] and the fact that diagnoses are typically not based on spirometry measurement but on self-reported symptoms [21]. Proposed reforms to improve the management of lung health in India have focused on supporting health promotion, early diagnosis, and disease management through several mechanisms, including strengthening primary care systems, improving access to technologies, drugs, and diagnostics, empowering patients and providers, and improving governance and accountability [21,22]. However, these campaigns have yet to be rolled out in a large-scale, national effort. Initial findings from a novel primary care program to address COPD in Kerala suggested that the program did not affect the number of hospital or emergency department visits but did lead to greater availability of spirometry and inhalers and lower demand for injectable drugs and nebulization due to better disease management [23]. Future campaigns should apply lessons learned from this initial effort to inform broader intervention programs targeting key areas including health system strengthening, providing low-cost and easily available technologies, drugs, and diagnostics, and improving the oversight and efficiency of designed programs.





**Fig 3. Patterns of symptoms and self-reported disease among those with and without obstructive lung disease among older adults in the Longitudinal Aging Study in India (N=31,103).** Panel [A] shows the proportion of participants with and without obstructive lung disease reporting respiratory symptoms. Panel [B] shows the proportion of participants with and without obstructive lung disease with self-reported chronic obstructive pulmonary disease (COPD), chronic bronchitis, asthma, or any of the three conditions. Panel [C] shows the proportion of participants with previously



diagnosed or previously undiagnosed obstructive lung disease who report symptoms. All proportions are weighted to account for unequal sampling probabilities and selection into the spirometry sample. Error bars show 95% confidence intervals. Asterisks show the statistical significance of comparisons (* = <0.05; ** = <0.01; *** = <0.001).

**Table 2. Mean forced expiratory volume in one second/forced vital capacity (FEV$_1$/FVC), prevalence of obstructive lung disease, and prevalence ratios (PR) for obstructive lung disease by various demographic and health-related risk factors among older adults in the Longitudinal Aging Study in India (N = 31,103). 95% confidence intervals are shown in parentheses. PRs are derived from Poisson regression models with robust variance. REF is used to denote reference categories. Model 1 adjusts for age and gender. Model 2 additionally adjusts for smoking status. BMI refers to body mass index.**

| | Mean FEV$_1$/FVC | Obstructive lung disease prevalence | Obstructive lung disease PR (unadjusted) | Obstructive lung disease PR (model 1) | Obstructive lung disease PR (model 2) |
|---|---|---|---|---|---|
| Gender | | | | | |
| Men | 77.7 (77.3–78.1) | 17.3 (16.1–18.5) | REF | REF | REF |
| Women | 80.3 (79.9–80.8) | 11.3 (10.2–12.5) | 0.65 (0.59–0.73) | 0.61 (0.55–0.68) | 0.74 (0.66–0.83) |
| Age group | | | | | |
| 45–54 | 80.1 (79.8–80.4) | 9.6 (8.7–10.5) | REF | REF | REF |
| 55–64 | 79.1 (78.7–79.5) | 14.2 (13.0–15.5) | 1.49 (1.33–1.66) | 1.57 (1.41–1.75) | 1.54 (1.39–1.72) |
| 65–74 | 77.5 (76.9–78.1) | 19.2 (17.3–21.2) | 2.00 (1.79–2.24) | 2.10 (1.88–2.35) | 2.08 (1.86–2.32) |
| 75+ | 77.0 (75.7–78.3) | 25.5 (21.4–30.0) | 2.66 (2.17–3.26) | 2.84 (2.34–3.46) | 2.80 (2.31–3.39) |
| Rural/urban | | | | | |
| Urban | 79.2 (78.8–79.7) | 13.8 (12.5–15.2) | REF | REF | REF |
| Rural | 78.8 (78.4–79.3) | 14.7 (13.6–15.9) | 1.07 (0.95–1.20) | 1.03 (0.92–1.16) | 0.98 (0.87–1.10) |
| Caste | | | | | |
| No caste or other caste | 78.5 (78.0–79.0) | 13.9 (12.7–15.3) | REF | REF | REF |
| Scheduled caste | 78.7 (78.1–79.3) | 16.8 (15.0–18.7) | 1.20 (1.05–1.37) | 1.23 (1.08–1.40) | 1.18 (1.04–1.34) |
| Scheduled tribe | 80.0 (79.1–81.0) | 11.8 (9.8–14.2) | 0.85 (0.70–1.03) | 0.87 (0.71–1.05) | 0.83 (0.69–1.01) |
| Other backward class | 79.2 (78.7–79.6) | 14.2 (12.9–15.6) | 1.02 (0.89–1.16) | 1.03 (0.91–1.16) | 1.03 (0.91–1.16) |
| Education | | | | | |
| No school | 79.2 (78.7–79.7) | 15.1 (13.8–16.4) | REF | REF | REF |
| Less than secondary school | 78.5 (78.1–78.9) | 14.9 (13.6–16.4) | 0.99 (0.89–1.11) | 0.93 (0.83–1.04) | 0.95 (0.86–1.06) |
| Secondary school and higher | 79.0 (78.6–79.3) | 12.8 (11.6–14.0) | 0.85 (0.76–0.95) | 0.80 (0.72–0.90) | 0.88 (0.78–0.98) |
| BMI category | | | | | |
| Normal | 78.6 (78.2–79.0) | 15.3 (14.1–16.6) | REF | REF | REF |
| Underweight | 77.1 (76.3–77.9) | 22.8 (20.3–25.4) | 1.49 (1.33–1.66) | 1.35 (1.21–1.51) | 1.30 (1.17–1.45) |
| Overweight | 79.7 (79.2–80.2) | 11.1 (9.6–12.7) | 0.72 (0.63–0.84) | 0.76 (0.66–0.88) | 0.78 (0.67–0.89) |
| Obese | 80.3 (79.9–80.6) | 9.7 (8.7–10.7) | 0.63 (0.56–0.71) | 0.71 (0.63–0.80) | 0.74 (0.66–0.83) |
| Smoking status | | | | | |
| Never | 79.7 (79.3–80.0) | 12.1 (11.2–13.0) | REF | REF | REF |
| Former | 75.8 (74.7–77.0) | 25.1 (21.2–29.6) | 2.08 (1.74–2.48) | 1.66 (1.40–1.97) | 1.66 (1.40–1.97) |
| Current | 76.4 (75.7–77.1) | 23.1 (20.7–25.6) | 1.91 (1.70–2.14) | 1.72 (1.53–1.93) | 1.72 (1.53–1.93) |
| Cooking fuel type | | | | | |
| Clean cooking fuel | 79.1 (78.7–79.4) | 13.7 (12.7–14.8) | REF | REF | REF |
| Unclean cooking fuel | 78.9 (78.3–79.4) | 15.2 (13.8–16.7) | 1.11 (0.99–1.24) | 1.09 (0.98–1.22) | 1.03 (0.92–1.15) |
| Tuberculosis | | | | | |
| No | 79.0 (78.6-79.3) | 14.4 (13.5-15.4) | REF | REF | REF |
| Yes | 78.7 (74.7–82.8) | 23.5 (14.6–35.6) | 1.64 (1.06-2.53) | 1.56 (0.97-2.50) | 1.62 (1.03-2.54) |

Management of risk factors is another important aspect of appropriate disease management and control. However, most existing evidence comes from high-income settings. Associations with age, smoking status, and tuberculosis are quite consistent across studies and were replicated in the LASI sample [24,25]. However, gender differences are less well understood [26]. Although earlier data from high-income settings largely also found evidence of lower risk among women, more recent data suggest a narrowing or reversal of this gender difference [27,28], potentially due to changes in smoking [26]. In our study, women had a lower risk of obstructive lung disease, and this gender difference persisted after adjusting for smoking status, but the adjustment was based on a crude classification of never vs. former vs. current smokers. Patterns in observed associations in the LASI sample also point to disparities by socioeconomic status (SES). We observed a lower risk of obstructive pulmonary disease among those with higher educational attainment and significant associations between BMI and obstructive pulmonary disease. Underweight BMI, which can be interpreted as a marker of low SES, was associated with higher risk. Counter to evidence from high-income settings [29], overweight and obesity were associated with a lower risk among the LASI cohort. However, in India, overweight and obesity are strongly associated with higher measures of SES, and protective associations have been observed across other health outcomes [30], indicating that benefits attributable to higher SES may offset potential negative health impacts of higher body mass in this setting. Although evidence suggests indoor pollution is an important risk factor for obstructive lung disease [31], in this study, the association between unclean cooking fuel use and obstructive lung disease was only marginally significant and was further attenuated after adjustment for smoking status. This finding may be due to the crude nature of the unclean cooking fuel indicator, which was self-reported and did not consider information on kitchen location, home ventilation, or non-cooking sources of indoor pollution. Future work should refine self-reported estimates and consider the collection of objective data on indoor and ambient air pollution, which was not included in the present analysis.

This study is the first nationally representative study of middle-aged and older adults in India to utilize objective spirometry data and report the national and regional prevalence of obstructive lung disease measured via spirometry. Importantly, use of objective measures of lung health from spirometry allows for the estimation of the prevalence of obstructive lung disease in a nationally representative sample without risk of bias due to under-reporting given low disease awareness.

However, limitations should be considered. First, due to the logistical complexity of administering a large population representative survey, spirometry was completed in participant's homes rather than a controlled medical setting, and no assessment of post-bronchodilator spirometry was included. However, review of the data included substantial attention to the quality of measurement, and those with unacceptable measurements were excluded. Lack of data on post-bronchodilator spirometry precluded the estimation of the prevalence of COPD specifically; instead, we focused on the prevalence of obstructive lung disease, including asthma and other conditions that cause a reduced $FEV_1/FVC$ ratio. However, estimates are still important for understanding the burden of lung disease and associated risk factors and for health resource planning and policy purposes. Second, selection bias due to the spirometry consent process, contraindications to spirometry, and challenges with spirometry measurement leading to unacceptable measurements were a concern. However, we developed inverse probability weights to account for selection bias and showed that the weighted sample did not differ from the full sample on important demographic and health-related characteristics. Selection bias due to pre-study mortality may have also impacted risk factor associations if risk factors were associated with death before study entry at age 45. Therefore, estimated associations should be interpreted as conditional on survival to 45 years of age. Third, absent Indian-specific parameters for the Global Lung Function Initiative prediction equations, we used Southeast Asian parameters to define severity groups. Future efforts should incorporate data from LASI into global models to improve available prediction equations. Fourth, we highlighted only a few selected risk factors in our analyses of associations and the extent to which the observed regional variability could be explained, but future work should explore a larger range of socioeconomic, environmental, occupational, and lifestyle factors that may influence lung disease risk.

The high prevalence of obstructive lung disease in adults 45 years and older based on spirometry calls attention to obstructive lung disease as an important public health concern in India. Despite geographic heterogeneity, prevalence estimates were greater than 10% in all regions, suggesting that nationwide campaigns are needed in addition to targeted regional efforts. Findings also call attention to low disease awareness, highlighting issues around diagnosis. Although we identified a few key demographic and risk factor correlates in this setting, future research is needed to improve estimates of associations. Research should consider a broader range of exposures and focus on isolating potentially causal effects, given the importance of risk factor management as a strategy to reduce disease prevalence. Ultimately, multifaceted, data-driven approaches are required to adequately address the burden of obstructive lung disease in India.

## Supporting information

**S1 Fig. Flowchart illustrating sample selection processes.**
(PDF)

**S1 File. Details on other measured variables included in analyses.**
(PDF)

**S1 Table. Distribution of body mass index for Indian-specific and WHO categories.**
(PDF)

**S2 Table. Prevalence ratios using Indian-specific and WHO thresholds.**
(PDF)

**S2 File. Details on covariates included in inverse probability weights.**
(PDF)

**S2 Fig. Standardized mean differences for inverse probability weights.**
(PDF)

**S3 Fig. Map depicting the categorization of regions with India.**
(PDF)

**S3 Table. Prevalence by geographic region.**
(PDF)

**S4 Table. Prevalence by age and gender.**
(PDF)

**S5 Table. Objective prevalence, self-reported prevalence, and disease awareness by region.**
(PDF)

**S4 Fig. Maps showing objective prevalence, self-reported prevalence, and disease awareness by region.**
(PDF)

**S6 Table. Prevalence comparisons between regions.**
(PDF)

## Acknowledgments

We thank all LASI team members and collaborators for their contribution to the Wave 1 LASI data collection.

## Author contributions

**Conceptualization:** T. V. Sekher, Emma Nichols, Jinkook Lee.

**Data curation:** Sarang P. Pedgaonkar.

**Formal analysis:** Emma Nichols.

**Funding acquisition:** T. V. Sekher, David E. Bloom, Jinkook Lee.

**Investigation:** Emma Nichols, Sundeep Salvi, Sarang P. Pedgaonkar, Crystal M. North, Sarah Petrosyan, Jinkook Lee.

**Methodology:** Emma Nichols.

**Project administration:** T. V. Sekher, Sarang P. Pedgaonkar, Sarah Petrosyan, David E. Bloom, Jinkook Lee.

**Supervision:** T. V. Sekher.

**Visualization:** Emma Nichols.

**Writing – original draft:** Emma Nichols.

**Writing – review & editing:** T. V. Sekher, Emma Nichols, Sundeep Salvi, Sarang P. Pedgaonkar, Crystal M. North, Sarah Petrosyan, David E. Bloom, Jinkook Lee.

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
