## [Decision Letter · Decision Letter 0]

29 Apr 2025

PONE-D-25-12176
The prevalence and correlates of obstructive lung disease among adults aged 45 and above in India: findings from the Longitudinal Aging Study in India
PLOS ONE

Dear Dr. Nichols,

Thank you for submitting your manuscript to PLOS ONE. After careful consideration, we feel that it has merit but does not fully meet PLOS ONE’s publication criteria as it currently stands. Therefore, we invite you to submit a revised version of the manuscript that addresses the points raised during the review process.

We look forward to receiving your revised manuscript.

Kind regards,

George Kuryan

Academic Editor

PLOS ONE

3. Thank you for uploading your study's underlying data set. Unfortunately, the repository you have noted in your Data Availability statement does not qualify as an acceptable data repository according to PLOS's standards.

Additional Editor Comments (if provided):

Reviewers' comments:

Reviewer's Responses to Questions

**Comments to the Author**

1. Is the manuscript technically sound, and do the data support the conclusions?

Reviewer #1: Yes

2. Has the statistical analysis been performed appropriately and rigorously? 

Reviewer #1: Yes

3. Have the authors made all data underlying the findings in their manuscript fully available?

Reviewer #1: No

4. Is the manuscript presented in an intelligible fashion and written in standard English?

Reviewer #1: Yes

5. Review Comments to the Author

Reviewer #1: The study is very interesting and does provide an insight into the prevalence and correlates of Obstructive Lung Disease in individuals aged 45 and above in India. However, the study does have certain limitations which the authors may kindly address

1) There could have been an inherent bias in the sample selection. The study included individuals more than 45 years of age , excluding younger individuals and those with other health profiles and hence the findings may not be generalised.

2) The authors themselves have pointed out that the awareness of the disease was very low, 12% in males and around 11% in females. Do they think that the results of prevalence may be skewed because of under reporting?

3) PFT was performed at homes , thereby the quality could have been affected. Also a post bronchodilator test was not performed which again encompasses other diseases like asthma under the broad label of OAD.

4) TB is an important cause of Obstructive Lung Disease and somehow, the association between these two has not been captured at all in this study.

5) The authors have mentioned that there is significant regional variability but , have not delved into the depths of the socioeconomic, environmental, access to healthcare which could influence this.

6)There is a limited scope of risk factors in this study. The authors have not taken into account the occupational exposure, environmental exposure, air quality and lifestyle choices which could have affected the study.

7) The study emphasises a lot on the need for improved health care infrastructure so far as the diagnosis and management of Obstructive lung diseases is concerned. But, how the authors envisage to do this is not mentioned.

Regards.

6. PLOS authors have the option to publish the peer review history of their article (what does this mean?). If published, this will include your full peer review and any attached files.

Reviewer #1: No

---

## [Author Response · Author response to Decision Letter 1]

22 May 2025

Dr. George Kuryan

05/15/2025

RE: PONE-D-25-12176

Dear Dr. Kuryan,

We appreciate the opportunity to address the Reviewers’ comments and revise our manuscript, PONE-D-25-12176, “The prevalence and correlates of obstructive lung disease among adults aged 45 and above in India: findings from the Longitudinal Aging Study in India.” Below, please find item-by-item responses to the Reviewers’ comments, which are included verbatim. All page and paragraph numbers refer to locations in the revised manuscript. Text in red indicates text that has been added. We believe these critiques and our responses have made for a stronger paper.

Thank you,

Emma Nichols, PhD

Center for Economic and Social Research

University of Southern California

JOURNAL REQUIREMENTS

Comment 1: Please ensure that your manuscript meets PLOS ONE's style requirements, including those for file naming.

Response 1: We have now made adjustments to ensure alignment with PLOS ONE’s style requirements.

Comment 2: Please include a complete copy of PLOS’ questionnaire on inclusivity in global research in your revised manuscript. Our policy for research in this area aims to improve transparency in the reporting of research performed outside of researchers’ own country or community. The policy applies to researchers who have travelled to a different country to conduct research, research with Indigenous populations or their lands, and research on cultural artefacts. The questionnaire can also be requested at the journal’s discretion for any other submissions, even if these conditions are not met. Please find more information on the policy and a link to download a blank copy of the questionnaire here: https://journals.plos.org/plosone/s/best-practices-in-research-reporting. Please upload a completed version of your questionnaire as Supporting Information when you resubmit your manuscript.

Response 2: We have now completed the questionnaire and include it as a supporting document.

Comment 3: Thank you for uploading your study's underlying data set. Unfortunately, the repository you have noted in your Data Availability statement does not qualify as an acceptable data repository according to PLOS's standards. At this time, please upload the minimal data set necessary to replicate your study's findings to a stable, public repository (such as figshare or Dryad) and provide us with the relevant URLs, DOIs, or accession numbers that may be used to access these data. For a list of recommended repositories and additional information on PLOS standards for data deposition, please see https://journals.plos.org/plosone/s/recommended-repositories.

Response 3: Unfortunately, due to legal and ethical reasons, we are unable to publish the data in a fully open-access repository. However, the data are available after a relatively brief and simple registration and agreement process. We have now updated text in the Data availability statement to reflect the data access restrictions and describe the process for accessing data:

“Longitudinal Aging Study in India (LASI) data used in this study, with the exception of spirometry data, are available after completion of a signed data access agreement form on the websites of the Gateway to Global Aging Data (https://g2aging.org/survey/get_data) and the International Institute for Population Sciences (https://www.iipsindia.ac.in/content/LASI-data). Spirometry data will be made available to researchers on request. Registration and data access forms are required under the Institutional Review Board’s human subjects research policies and requirements.”

Comment 4: Please include a separate caption for each figure in your manuscript.

Response 4: We have separate captions for each figure in the manuscript, which now appear in the manuscript text following the paragraph at first mention.

Comment 5: Please review your reference list to ensure that it is complete and correct. If you have cited papers that have been retracted, please include the rationale for doing so in the manuscript text or remove these references and replace them with relevant current references. Any changes to the reference list should be mentioned in the rebuttal letter that accompanies your revised manuscript. If you need to cite a retracted article, indicate the article’s retracted status in the References list and also include a citation and full reference for the retraction notice.

Response 5: We have now reviewed all included citations for completeness and accuracy.

REVIEWER 1

Comment 1: The study is very interesting and does provide an insight into the prevalence and correlates of Obstructive Lung Disease in individuals aged 45 and above in India.

Response 1: We thank the Reviewer for this comment.

Comment 2: There could have been an inherent bias in the sample selection. The study included individuals more than 45 years of age , excluding younger individuals and those with other health profiles and hence the findings may not be generalised.

Response 2: The LASI sample is representative of those over 45 years of age. Prevalence estimates should not be biased by this age restriction, but rather results are only applicable to this adult population and are not relevant for younger age groups. We have added text in several places to make this age restriction clearer in the interpretation of findings. This text now reads:

Introduction (Lines 47-48):

“The Longitudinal Aging Study in India (LASI) is, to our knowledge, the first study to administer objective spirometry assessments to a nationally representative sample of adults (over 45 years of age) in India.”

Discussion (Lines 310-311):

“The high prevalence of obstructive lung disease in adults 45 years and older based on spirometry calls attention to obstructive lung disease as an important public health concern in India.”

Given the study began at age 45, survival bias may impact associations between risk factors and lung health outcomes if risk factors were associated with pre-study mortality. Therefore, results of risk factor analyses can be interpreted as conditional on survival to age 45. We have added text to the limitations section to acknowledge this interpretation and make this clear to readers (Lines 300-303):

“Selection bias due to pre-study mortality may have also impacted risk factor associations if risk factors were associated with death before study entry at age 45. Therefore, estimated associations should be interpreted as conditional on survival to 45 years of age.”

Further, exclusion criteria were not based on health profiles. However, we were concerned that exclusions due to lack of consent to spirometry or unacceptable spirometry measurements could induce bias. Therefore, we estimated inverse probability weights to ensure that those sample exclusions did not lead to differences on observable characteristics between the full sample and the final analytic sample used in the paper. In addition to the previously included text describing the potential for selection bias and implemented solution in the discussion (shown below), we further added text to the methods to highlight this analysis when these sample selections are first mentioned.

Methods (Lines 90-91):

“To account for potential selection bias due to lack of consent or inacceptable spirometry measurements, we used inverse probability weights (see below).”

Discussion (Lines 296-300):

“Second, selection bias due to the spirometry consent process, contraindications to spirometry, and challenges with spirometry measurement leading to unacceptable measurements were a concern. However, we developed inverse probability weights to account for selection bias and showed that the weighted sample did not differ from the full sample on important demographic and health-related characteristics.”

Comment 3: The authors themselves have pointed out that the awareness of the disease was very low, 12% in males and around 11% in females. Do they think that the results of prevalence may be skewed because of under reporting?

Response 3: One of the strengths of using objective assessments of lung health based on spirometry is that our estimates should not be impacted by potential under-reporting or lack of awareness. All prevalence estimates that we provide are based on objective spirometry assessments to ensure that this under-reporting does not influence the findings of the present paper. We are then able to estimate awareness by directly comparing the objective measure to the self-report measure. To make sure this clear to readers, we have now added some text explicitly discussing this point (Lines 285-287):

“Importantly, use of objective measures of lung health from spirometry allows for the estimation of the prevalence of obstructive lung disease in a nationally representative sample without risk of bias due to under-reporting given low disease awareness.”

Comment 4: PFT was performed at homes , thereby the quality could have been affected. Also a post bronchodilator test was not performed which again encompasses other diseases like asthma under the broad label of OAD.

Response 4: We agree that lack of post-bronchodilator spirometry test, and potential concerns about the quality of spirometry assessments are notable limitations of the current study. However, these tradeoffs were necessary to ensure feasibility of data collection during

fieldwork in this large, representative sample. We did pay careful attention to the quality of the measurement: field investigators who conducted spirometry underwent a 5-day rigorous and standardized training protocol, and spirometry experts from the Chest Research Foundation in Pune reviewed all data for American Thoracic Society/European Respiratory Society acceptability. Though we previously described how spirometry experts reviewed the data quality, we have now also added text on the training protocol to the methods section (Lines 72-74):

“Interviewers underwent a 5-day rigorous and standardized training protocol. Following training, interviewers measured forced expiratory volume in one second (FEV1) and forced vital capacity (FVC) with Thor handheld ultrasonic spirometers (Thor Medical Systems, Budapest, Hungary).”

Despite limitations, given the uniqueness of this data in a large, nationally representative study in India, we feel that that the information obtained from pre-bronchodilator spirometry conducted in the home still represents a significant contribution to the knowledge of lung disease in India and to the scientific literature. However, we understand the importance of acknowledging these limitations and their implications. We have now updated the included text in the limitations section to address this topic (Lines 288-296):

“First, due to the logistical complexity of administering a large population representative survey, spirometry was completed in participant’s homes rather than a controlled medical setting, and no assessment of post-bronchodilator spirometry was included. However, review of the data included substantial attention to the quality of measurement, and those with unacceptable measurements were excluded. Lack of data on post-bronchodilator spirometry precluded the estimation of the prevalence of COPD specifically; instead, we focused on the prevalence of obstructive lung disease, including asthma and other conditions that cause a reduced FEV1/FVC ratio. However, estimates are still important for understanding the burden of lung disease and associated risk factors and for health resource planning and policy purposes.”

Comment 5: TB is an important cause of Obstructive Lung Disease and somehow, the association between these two has not been captured at all in this study.

Response 5: We thank the Reviewer for this suggestion. The LASI survey includes a measure of self-reported tuberculosis (TB) in the past two years. We have now included this throughout the paper. Though the prevalence of self-reported TB in the past two years is low, we did observe associations between TB and lung disease in the current sample, and we believe this is a useful addition to the paper. Please see below some of the added text:

Methods (Lines 100-102):

“Covariates included respondents’ self-reported age, gender, education, literacy, marital status, caste, smoking status, frequency of moderate and vigorous physical activity, use of unclean cooking fuel, and self-reported tuberculosis in the last two years.”

Results (Lines 202-204):

“In contrast, older age, scheduled caste (compared with no caste or other caste), underweight BMI, smoking, unclean fuel use, and self-reported tuberculosis were associated with a higher risk of obstructive lung disease.”

Discussion (Lines 260-262):

“Associations with age, smoking status, and tuberculosis are quite consistent across studies and were replicated in the LASI sample [24,25].”

Please see the updated manuscript for edits to Tables 1 & 2.

Comment 6: The authors have mentioned that there is significant regional variability but , have not delved into the depths of the socioeconomic, environmental, access to healthcare which could influence this.

Response 6: The goals of the current paper were to establish prevalence estimates for obstructive lung disease in India overall as well as by age and gender, assess disease awareness, and evaluate associations with several key risk factors. We agree that the exploration of a larger range of risk factors and the role of these risk factors in explaining regional variability is a worthy topic, but this was beyond the scope of this current paper. We now include additional text calling attention to this important research area and suggest that future work should more thoroughly investigate this topic. This text reads (Lines 306-309):

“Fourth, we highlighted only a few selected risk factors in our analyses of associations and the extent to which the observed regional variability could be explained, but future work should explore a larger range of socioeconomic, environmental, occupational, and lifestyle factors that may influence lung disease risk.”

Comment 7: There is a limited scope of risk factors in this study. The authors have not taken into account the occupational exposure, environmental exposure, air quality and lifestyle choices which could have affected the study.

Response 7: The evaluation of a broader range of risk factors is a worthy topic but was beyond the scope of the present paper. Please see Response 6 above for added text to acknowledge this important line of research.

Comment 8: The study emphasises a lot on the need for improved health care infrastructure so far as the diagnosis and management of Obstructive lung diseases is concerned. But, how the authors envisage to do this is not mentioned.

Response 8: We have now expanded on text in the Discussion section to be more specific in discussing the different components of proposed reforms and the key areas that should be targeted in efforts to improve the diagnosis and management of lung disease. This text now reads (Lines 248-258):

“Proposed reforms to improve the management of lung health in India have focused on supporting health promotion, early diagnosis, and disease management through several mechanisms, including strengthening primary care systems, improving access to technologies, drugs, and diagnostics, empowering patients and providers, and improving governance and accountability [21,22]. However, these campaigns have yet to be rolled out in a large-scale, national effort. Initial findings from a novel primary care program to address COPD in Kerala suggested that the program did not affect the number of hospital or emergency department visits but did lead to greater availability of spirometry and inhalers and lower demand for injectable drugs and nebulization due to better disease management [23]. Future campaigns should apply lessons learned from this initial effort to inform broader intervention programs targeting key areas including health system strengthening, providing low-cost and easily available technologies, drugs, and diagnostics, and improving the oversight and efficiency of designed programs.”

---

## [Decision Letter · Decision Letter 1]

16 Jun 2025

The prevalence and correlates of obstructive lung disease among adults aged 45 and above in India: findings from the Longitudinal Aging Study in India

PONE-D-25-12176R1

Dear Dr. Nichols

We’re pleased to inform you that your manuscript has been judged scientifically suitable for publication and will be formally accepted for publication once it meets all outstanding technical requirements.

Kind regards,

George Kuryan

Academic Editor

PLOS ONE

Additional Editor Comments (optional):

Reviewers' comments:

Reviewer's Responses to Questions

**Comments to the Author**

1. If the authors have adequately addressed your comments raised in a previous round of review and you feel that this manuscript is now acceptable for publication, you may indicate that here to bypass the “Comments to the Author” section, enter your conflict of interest statement in the “Confidential to Editor” section, and submit your "Accept" recommendation.

Reviewer #1: All comments have been addressed

2. Is the manuscript technically sound, and do the data support the conclusions?

Reviewer #1: Yes

3. Has the statistical analysis been performed appropriately and rigorously? 

Reviewer #1: Yes

4. Have the authors made all data underlying the findings in their manuscript fully available?

Reviewer #1: Yes

5. Is the manuscript presented in an intelligible fashion and written in standard English?

Reviewer #1: Yes

6. Review Comments to the Author

Reviewer #1: The authors have addressed all comments to satisfaction and most of the limitations have been added in the discussion as well.

7. PLOS authors have the option to publish the peer review history of their article (what does this mean?). If published, this will include your full peer review and any attached files.

Reviewer #1: No

---

## [Editor Report · Acceptance letter]

PONE-D-25-12176R1

PLOS ONE

Dear Dr. Nichols,

I'm pleased to inform you that your manuscript has been deemed suitable for publication in PLOS ONE. Congratulations! Your manuscript is now being handed over to our production team.

Kind regards,

on behalf of

Professor George Kuryan

Academic Editor

PLOS ONE